# *Spiraea prunifolia* var. *simpliciflora* Attenuates Oxidative Stress and Inflammatory Responses in a Murine Model of Lipopolysaccharide-Induced Acute Lung Injury and TNF-α-Stimulated NCI-H292 Cells

**DOI:** 10.3390/antiox9030198

**Published:** 2020-02-26

**Authors:** Ba-Wool Lee, Ji-Hye Ha, Han-Gyo Shin, Seong-Hun Jeong, Da-Bin Jeon, Ju-Hong Kim, Ji-Young Park, Hyung-Jun Kwon, Kyungsook Jung, Woo-Song Lee, Hyeon-Young Kim, Sung-Hwan Kim, Hyun-Jae Jang, Young-Bae Ryu, In-Chul Lee

**Affiliations:** 1Functional Biomaterial Research Center, Korea Research Institute of Bioscience and Biotechnology, Jeongeup-si, Jeollabuk-do 56212, Korea; pl0706@kribb.re.kr (B.-W.L.); jihye2640@kribb.re.kr (J.-H.H.); shangyo48@naver.com (H.-G.S.); jsh0830@kribb.re.kr (S.-H.J.); wjsekqls4125@gmail.com (D.-B.J.); wnghd93@kribb.re.kr (J.-H.K.); loveme@kribb.re.kr (J.-Y.P.); hjkwon@kribb.re.kr (H.-J.K.); jungks@kribb.re.kr (K.J.); wslee@kribb.re.kr (W.-S.L.); 2National Center for Efficacy Evaluation of Respiratory Disease Product, Korea Institute of Toxicology, Jeongeup-si, Jeollabuk-do 56212, Korea; hyeonyoung.kim@kitox.re.kr (H.-Y.K.); sunghwan.kim@kitox.re.kr (S.-H.K.); 3Natural Medicine Research Center, Korea Research Institute of Bioscience and Biotechnology, Cheongju-si, Chungcheongbuk-do 28116, Korea; water815@kribb.re.kr

**Keywords:** *Spiraea prunifolia* var. *simpliciflora*, acute lung injury, nuclear factor-kappaB, anti-oxidant activity, nuclear factor erythroid 2-related factor

## Abstract

*Spiraea prunifolia* var. *simpliciflora* (SP) is traditionally used as an herbal remedy to treat fever, malaria, and emesis. This study aimed to evaluate the anti-oxidative and anti-inflammatory properties of the methanol extract of SP leaves in tumor necrosis factor (TNF)-α-stimulated NCI-H292 cells and in a lipopolysaccharide (LPS)-induced acute lung injury (ALI) mouse model. SP decreased the number of inflammatory cells and the levels of TNF-α, interleukin (IL)-1β, and IL-6 in the bronchoalveolar lavage fluid, and inflammatory cell infiltration in the lung tissues of SP-treated mice. In addition, SP significantly suppressed the mRNA and protein levels of TNF-α, IL-1β, and IL-6 in TNF-α-stimulated NCI-H292 cells. SP significantly suppressed the phosphorylation of the mitogen-activated protein kinases (MAPKs) and p65-nuclear factor-kappa B (NF-κB) in LPS-induced ALI mice and TNF-α-stimulated NCI-H292 cells. SP treatment enhanced the nuclear translocation of nuclear factor erythroid 2-related factor (Nrf2) with upregulated antioxidant enzymes and suppressed reactive oxygen species (ROS)-mediated oxidative stress in the lung tissues of LPS-induced ALI model and TNF-α-stimulated NCI-H292 cells. Collectively, SP effectively inhibited airway inflammation and ROS-mediated oxidative stress, which was closely related to its ability to induce activation of Nrf2 and inhibit the phosphorylation of MAPKs and NF-κB. These findings suggest that SP has therapeutic potential for the treatment of ALI.

## 1. Introduction

Acute lung injury (ALI) is a severe respiratory disorder that underlies acute and persistent lung inflammation [1,2]. It is characterized by neutrophil influx, alveolar-capillary barrier damage leading to interstitial edema, and the impairment of respiratory function [3,4]. While great advances in understanding the pathophysiology of ALI have been reported to date, no effective medicine is available for ALI, and it remains the leading cause of morbidity and mortality in critically ill patients [5,6].

Lipopolysaccharide (LPS), from the outer membrane of gram-negative bacteria, enhances oxidative stress by generating of reactive oxygen species (ROS) and an inflammatory response in an experimental animal model of ALI [7,8,9,10]. LPS is recognized by toll-like receptor 4 (TLR4), which activates downstream signaling via nuclear factor-kappa B (NF-κB) and mitogen-activated protein kinase (MAPK) pathways [11]. The activation of NF-κB and MAPK pathways collaborates to induce the release of various pro-inflammatory cytokines or chemokines, including tumor necrosis factor (TNF)-α, interleukin (IL)-1β, and IL-6 [12,13]. Oxidative stress also plays a major role in the pathogenesis of lung injury in LPS-induced ALI [14,15]. Nuclear factor erythroid-2 related factor 2 (Nrf2) is a redox-sensitive transcription factor and it regulates the antioxidant and detoxifying enzymes [16]. It has been reported that Nrf2 plays an essential role in protecting cells against inflammatory and oxidative stress-mediated diseases [17,18]. Under normal conditions, Nrf2 binds to Kelch-like ECH-associated protein 1 (Keap1) in the cytoplasm. Upon activation, Nrf2 dissociates from Keap1 and translocates into the nucleus, where it binds to antioxidant response elements (AREs) in the promoter region of cytoprotective antioxidant enzymes, such as NAD(P)H quinone dehydrogenase 1 (NQO1) and heme oxygenase-1 (HO-1) [17]. Emerging evidence indicates that Nrf2 activation plays a protective role against LPS-induced oxidative stress and inflammation in the LPS-induced ALI model [14,18]. Recently, many researchers have begun to investigate the effects of herbal-derived natural products with anti-oxidant or anti-inflammatory properties on ALI [7,19].

*Spiraea prunifolia* var. *simpliciflora* (SP), which is a member of the Rosaceae family, grows throughout northeast Asia. The young leaves, fruits, and roots of SP have been used as herbal remedies for malaria, fever, and emesis [20,21,22]. Previous studies demonstrated that a methanol extract of SP root had potent antioxidative and anti-inflammatory effects in LPS-stimulated RAW264.7 cells [21,22]. In addition, SP extract exhibited radical-scavenging activity and inhibited nitric oxide (NO) production [21,23]. However, there is no evidence for the anti-inflammatory effects of SP in ALI in vivo. Therefore, we evaluated the effects of a methanol extract of SP leaves on inflammation in TNF-α-stimulated human airway epithelial (NCI-H292) cells and in an LPS-induced ALI mouse model. 

## 2. Materials and Methods

### 2.1. UPLC Q-TOF/MS Analysis

The methanol extract from the SP leaves was obtained from The Korea Plant Extract Bank of Korea Research Institute of Bioscience and Biotechnology (KRIBB, PB3132.8). SP belongs to the Rosaceae family and it is collected from the Republic of Korea (Chungcheongnam-do, April). The metabolomics analysis of SP leaves was chromatographically analyzed by ACQUITY UPLC system that was coupled with Vion IMS QToF mass spectrometer (Waters Corp., Milford, MA, USA) while using BEH C18 column (2.1 × 100 mm, 1.7 μm) and two mobile phases, 0.1% formic acid in water (A) and acetonitrile (B). The column and sample tray temperature were maintained at 35 °C and 10 °C, respectively. The flow rate was 0.4 mL/min. and elution conditions were optimized as follows: 0–2 min., 5% B; 2–12 min., 5–30% B; and, 12–15 min., 30–100% B. The mass spectrometer operated in negative mode from 100 to 1500 Da with a 0.2 s scan time while using a desolvation temperature of 350 °C, source temperature of 110 °C, and cone voltage of 40 V. Accurate mass data were corrected during acquisition while using an external reference (Lock-Spray^TM^), which generated a reference ion of leucine encephalin (50 pg/mL) at 556.2771.

### 2.2. Animal Husbandry

Specific pathogen-free male C57BL/6 mice (20–25 g, 6–7 weeks old) were purchased from Orient Bio (Seongnam, Republic of Korea). They were housed in groups of four under standard conditions (temperature 22 ± 2 °C, humidity 55 ± 5%, 12-h light/dark cycle) with food and water. The Institutional Animal Care and Use Committee of the Korea Research Institute of Bioscience and Biotechnology approved all procedures (Approved number: KRIBB-AEC-18205).

### 2.3. LPS-induced ALI Model and Differential Cell Count in Bronchoalveolar Lavage Fluid (BALF) Collection

The SP (50 or 100 mg/kg) was dissolved in distilled water with 2% dimethyl sulfoxide (DMSO; Sigma-Aldrich, St. Louis, MO, USA) and dexamethasone (DEX 3 mg/kg) were dissolved in distilled water before treatment daily. SP and DEX were orally administered from day 0 to day 5. DEX was used as a positive control [7]. The mice were treated with LPS (form *Escherichia coli* O111:B4; Sigma-Aldrich) 20 μg in 50 μL phosphate-buffered saline (PBS; Gibco, San Diego, CA, USA) by intranasal (i.n.) instillation 1 h after SP and DEX treatment on day 3. The NC group was treated with vehicle (2% DMSO) and given 50 μL PBS only by i.n. instillation on day 3. It has been reported that the male mice were more susceptible to LPS-induced airway inflammation as compared to female mice [24]. Thus, we used only male mice in the LPS-induced ALI model. A total of 35 male mice were randomly divided into the control and four treatment groups. The animals were housed three or four per cage and each group consisted of seven mice.

Normal control (NC) group: treated with vehicle (2% DMSO) from day 0 to day 5 and given 50 μL PBS without LPS on day 3LPS group: treated with vehicle (2% DMSO) from day 0 to day 5 and given LPS 20 μg in 50 μL PBS on day 3DEX group: treated with DEX 3 mg/kg only from day 0 to day 5 and given LPS 20 μg in 50 μL PBS on day 3SP50 group: treated with SP 50 mg/kg only and given LPS 20 μg in 50 μL PBS on day 3.SP100 group: treated with SP 100 mg/kg only and given LPS 20 μg in 50 μL PBS on day 3.

BALF collection was performed while using the method of Shin et al. [7]. Briefly, ice-cold PBS (0.7 mL) was infused into the lungs twice and withdrawn each time using a tracheal cannula to obtain the BALF (a total volume of 1.4 mL). The collected BALF was centrifuged at 1000 rpm for 10 min. at 4 °C. The supernatants were collected and then stored at −70 °C before cytokine analysis. The cell pellet was re-suspended with 1 mL ice-cold PBS and then attached on the slide while using the Cytospin 4 centrifuge (1000 rpm, 5 min., 20 °C). The differential cell count was performed using Diff-Quik^®^ staining reagent (IMEB, San Marcos, CA, USA), according to the manufacturer’s instructions. Five images of each slide were taken by Leica DM5000B microscope while using Leica Application Suite acquisition software (Leica Microsystems, Wetzlar, Germany) under 40× objective lens. The total cells, neutrophils, macrophages, and other cells were counted in a double-blinded manner.

### 2.4. Measurement of Cytokines and Protein Contents in BALF

Pro-inflammatory cytokines, TNF-α, IL-1β, and IL-6, in the BALF were measured while using enzyme-linked immunosorbent assay (ELISA) kits (R&D system, Minneapolis, MN, USA) according to the manufacturer’s instructions. The absorbance of each sample was measured at 450 nm in a microplate reader (iMark^TM^, Bio-Rad Laboratories, Richmond, CA, USA). The determination of cytokine level was performed in triplicate for each sample (*n* = 7/group). The absolute concentrations were calculated by running standard curves on identical ELISA plates.

The protein contents in the supernatants of the BALF were quantified while using a Bradford reagent to evaluate vascular permeability to airways. Briefly, the 10 μL of the supernatants were placed in a flat-bottom 96-well ELISA plate, and 200 μL of Quick Start^TM^ Bradford 1× Dye Reagent (Bio-Rad Laboratories) was added and incubated at 20 °C for 10 min. Absorbance was measured at 595 nm on a microplate reader (Bio-Rad Laboratories). The determination of protein content was performed in duplicate (*n* = 7/group). A standard curve was established while using a serial dilution of protein standard.

### 2.5. Lung Tissue Histopathology

A portion of the lung tissue was dissected and fixed in 10% neutral buffered formalin solution. The fixed tissues were embedded in paraffin, sectioned to 4 μm thickness, deparaffinized, and then rehydrated. The extent of lung injury was evaluated by staining with hematoxylin and eosin (H&E). All of the lesions were manually examined with a light microscope (Leica Microsystems) with 10× and 20× objective lens in a totally blind manner for each slide (*n* = 7/group). The main histological lesions, leukocyte infiltration and thickening of alveolar walls, were graded as follows: 0, no lesions; 1, mild; 2, moderate; and, 3, severe.

### 2.6. Immunoblotting

The lung tissues were homogenized (1/10, w/v) in a T-PER Tissue Protein Extraction Reagent (Thermo Scientific, Waltham, MA, USA) containing protease and phosphatase inhibitor cocktail (Thermo Scientific). The protein concentration for each sample was determined while using a Bradford reagent (Bio-Rad Laboratories, Richmond, CA, USA). Equal amounts of total protein (30 μg) were resolved by 4–12% SDS-polyacrylamide gel electrophoresis and then transferred to polyvinyl difluoride membranes. The membranes were incubated with blocking solution (Thermo Scientific) followed by overnight incubation at 4°C with the following primary antibodies and dilutions: Erk, p-Erk, JNK, p-JNK, p38MAPK, p-p38MAPK, p65NF-κB, p-p65NF-κB, β-actin (1:1000 dilution; Cell Signaling Technology, Danvers, MA, USA), inducible nitric oxide synthase (iNOS), Nrf2, HO-1, NQO1, and TNF-α (1:1000 dilution; Abcam, Cambridge, UK). The membranes were washed three times with Tris-buffered saline containing 0.05% Tween 20 (TBST), followed by incubation with a 1:10000 dilution of horseradish peroxidase-conjugated secondary antibody (Cell Signaling Technology, Danvers, MA, USA) for 1 h at room temperature. The blots were washed again three times with TBST. The protein bands were developed while using an enhanced chemiluminescence kit (Thermo Scientific). The protein expression for each sample was measured in duplicate (*n* = 7/group). Densitometric analysis for each protein band was determined using chemiluminescent scanner (LI-COR, Biosciences, Lincoln, NE, USA).

### 2.7. Oxidative Stress Markers Analysis

The weighed lung tissues were grinded with glass beads with ice-cold PBS (pH 7.4) to obtain 1:9 (w/v) whole homogenate. The homogenates were centrifuged at 11,000 × *g* for 15 min. at 4 °C to discard cell debris. The collected supernatant was used for the measurement to oxidative stress levels by determining the contents of thiobarbituric acid-reactive substances (TBARS), which is a marker of lipid peroxidation, and glutathione (GSH) while using a commercial assay kit (DoGen), according to manufacturer’s instruction. The levels of GSH and TBARS were determined in triplicate for each sample (*n* = 7/group).

In brief, 200 μL tissue lysates was mixed with 500 μL trichloroacetic acid and incubated at 10 min. at 4 °C to determine TBARS. After incubation, the samples were centrifuged at 3000× *g* for 10 min. at 4 °C. The supernatant was collected and mixed with 0.025 M TBA and incubated for 45 min. at 65 °C. The absorbance of samples was determined at 540 nm while using microplate readers and converted to μmols using a standard curve (range 0–20 μM). The TBASRS level in the sample was expressed as nmol/mg protein.

50 μL lung tissue lysates were mixed with 350 μL 5% metaphosphoric acid and then vortexed for 20 s. to determine the GSH concentration. The samples were centrifuged at 3000× *g* for 10 min. at 4 °C. The supernatants were mixed with 5,5’-dithiobis-2-nitrobenzoid acid (DTNB), as a chromogen, and added glutathione reductase, and incubated for 5 min. at room temperature. After incubation, nicotinamide adenine dinucleotide phosphate (NADPH) was added in the samples and the absorbance was determined at 415 nm while using microplate readers. The contents of GSH was calculated by extrapolation on a standard curve and expressed nmol/mg protein.

### 2.8. Effects of SP on ROS Production and 2,2-Diphenyl-1-Picryl Hydrazyl (DPPH) Radical Scavenging Activity

After removing the medium, the cells were used to determine intracellular ROS generation while using the fluorescent dye 2’,7’-Dichlorofluorescin diacetate (DCFDA) cellular ROS detection assay kit (Abcam), according to manufacturer’s instruction. DCFDA assay was performed in triplicate for each sample (*n* = 3). Briefly, the cells were washed with buffer and incubated with 25 μM DCFDA at 37 °C for 1 h. 50 μM of tert-butyl hydrogen peroxide (TBHP) was used as ROS positive control. After incubation, fluorescence intensity was determined by the excitation and emission = 485/535 nm.

The DPPH (Sigma-Aldrich, St. Louis, MO, USA) radical scavenging activity of SP was measured according to Lim et al. [19]. 200 μM DPPH solution (dissolved in methanol, 100 μL) and the same volume of diluted extract (0, 1.56, 3.12, 6.25, 12.5, 25, 50, and 100 μg/mL) were mixed in 96-well microplate and reactive at room temperature for 30 min. The DPPH radical scavenging activity for each sample (*n* = 3) was determined in triplicate. The absorbance was measured at 520 nm with a blank containing DPPH and methanol. The DPPH radical scavenging activity was calculated according to the equation.
DPPH radical scavenging activity (inhibition %) = (A_sample_ − A_sample blank_)/(A_DPPH control_ − A_solvent control_) ×100

### 2.9. Cell Culture and Cell Viability

The human lung epithelial cell line NCI-H292 cell was purchased from ATCC (CRL-1848, Manassas, VA, USA). The NCI-H292 cells were maintained in RPMI 1640 media (Gibco) that was supplemented with 10% heat-inactivated fetal bovine serum (FBS; Gibco, San Diego, CA, USA) and antibiotics at 37 °C in a 5% CO_2_ incubator.

The NCI-H292 cells were seeded in 96-well plates at a density of 5 × 10^4^ cells/well and incubated in 0.1% FBS medium and treated with various concentrations of SP (0, 10, 25, 50, and 100 μg/mL). After 24 h incubation, the effect of SP on the cell viability was measured while using water-soluble tetrazolium salt-1 (WST-1; EZ-CyTox, DoGen, Republic of Korea), according to the manufacturer’s instructions. 10% total volume of WST-1 solution was added to each well and then incubated for 3 h. The cell viability was determined in triplicate for each concentration (*n* = 3). The absorbance was measured while using a microplate reader (Bio-Rad Laboratories, Richmond, CA, USA) at 450 nm. The optical density of control cells (untreated) was taken as 100% viability.

### 2.10. Effects of SP on Pro-Inflammatory Cytokine Levels and Oxidative Stress Markers in TNF-α-stimulated NCI-H292 Cells

The cells (5 × 10^4^ cells/well) were incubated with SP at 0, 25, 50, and 100 μg/mL for 1 h before human recombinant TNF-α 30 ng/mL (Peprotech, Rocky Hill, NJ, USA) treatment. The levels of TNF-α and IL-6 in culture medium were determined at 30 min. and 24 h incubation, respectively, after TNF-α treatment and then quantified while using a competitive ELISA kit (R&D system, Minneapolis, MN, USA), according to the manufacturer’s instructions. The absorbance at 450 nm was measured in a microplate reader (Bio-Rad Laboratories). After the collection of culture medium, cells were harvested and used to analyze the oxidative stress makers, TBARS and GSH levels, as described above. The levels of cytokines and oxidative stress markers were determined in triplicate for each sample (*n* = 3).

### 2.11. Effects of SP on MAPKs and NF-κB in TNF-α-stimulated NCI-H292 Cells

The cells (5 × 10^4^ cells/well) were incubated with SP (0, 25, 50, and 100 μg/mL) for 1 h, followed by incubation in the presence of TNF-α (30 ng/mL). The cells were incubated for 30 min., 2 h, and 24 h to determine the protein levels of MAPKs, NF-κB, and TNF-α, respectively. The cells were washed with PBS and collected with M-PER^TM^ Mammalian Protein Extraction Reagent (Thermo Scientific, Waltham, MA, USA) containing protease and phosphatase inhibitor cocktail. The protein concentration for each samples were determined while using a Bradford reagent (Bio-Rad Laboratories, Richmond, CA, USA). The expression levels of Erk, p-Erk, JNK, p-JNK, p38MAPK, p-p38MAPK, p65NF-κB, p-p65NF-κB (1:1000 dilution; Cell Signaling Technology, Danvers, MA, USA), and Nrf-2, HO-1, and TNF-α (1:1000 dilution; Abcam), and β-actin (loading control) were determined in triplicate (*n* = 3) while using immunoblotting, as described above.

### 2.12. Effects of SP on Nrf2 and HO-1 in TNF-α-Stimulated NCI-H292 Cells

The cells (5 × 10^4^ cells/well) were incubated with SP (0, 25, 50, and 100 μg/mL) for 1 h, followed by incubation in the presence of TNF-α (30 ng/mL). After 1 h incubation, the cells were washed with PBS (Gibco, San Diego, CA, USA). The cytoplasmic and nuclear fraction of cell was separated by using NE-PER Nuclear and Cytoplasmic Extraction Reagents (Thermo Scientific, Waltham, MA, USA) containing protease and phosphatase inhibitor cocktail. The protein concentration for each samples were determined while using a Bradford reagent (Bio-Rad Laboratories, Richmond, CA, USA). The expression levels of Nrf2, HO-1, and β-actin (loading control) were determined in triplicate (*n* = 3) while using immunoblotting, as described above.

### 2.13. Quantitative Real-Time Polymerase Chain Reaction (PCR)

The cells (5 × 10^4^ cells/well) were incubated with SP (0, 25, 50, and 100 μg/mL), followed by incubation in the presence of TNF-α (30 ng/mL) for 18 h. After incubation, the cells were washed twice with PBS and the total RNA was isolated using the RNA extraction kit (RNeasy^®^, Qiagen, Valencia, CA, USA). For real-time PCR, single-strand cDNA was synthesized from 1 μg total RNA. Real-time PCR was conducted in tri-plicate with the CFX96 Touch^TM^ Real-time PCR detection system (Bio-Rad Laboratories, Richmond, CA, USA), following the manufacture’s specifications, and SensiFast^TM^ SYBR No-ROX kit (Bionline, Tauton, MA, USA) was used to prepare the PCR reaction substrate. *TNF-α* forward, 5’-GAA AGC ATG ATC CGG GAC GTG-3’, *TNF-α* reverse, 5’-GAT GGC AGA GAG GAG GTT GAC-3’ (Gene-Bank accession number: NM_000594); *IL-6* forward, 5’-ATG CAA TAA CCA CCC CTG AC-3’, *IL-6* reverse, 5’-ATC TGA GGT GCC CAT GCT AC-3’ (Gene-Bank accession number: NM_000600); *IL-1β* forward, 5’-AGC CAG GAC AGT CAG CTC TC-3’, *IL-1β* reverse, 5’-ACT TCT TGC CCC CTT TGA AT-3’ (Gene-Bank accession number: NM_000576); *GAPDH* forward, 5’-CAA AAG GGT CAT CAT CTC TG-3’, *GAPDH* reverse, 5’-CCT GCT TCA CCA CCT TCT TG-3’ (Gene-Bank accession number: NM_002046). The samples (*n* = 3) were run in triplicate to ensure amplification integrity and standard PCR conditions were 95 °C for 3 min., 50 °C for 20 s, and 72 °C for 20 s, as recommended by the primer manufacturer. The threshold cycle (Ct; the cycle number at which the amount of the amplified gene of interest reaches a fixed threshold) was subsequently determined. The relative quantitation of each mRNA expression was calculated by the comparative Ct method. The relative quantitation values of targets were normalized to the endogenous GAPDH control gene and were expressed as 2^−ΔΔCt^ (fold), where ^Δ^Ct = Ct of target gene—Ct of endogenous control gene and ^ΔΔ^Ct = ^Δ^Ct of samples for target gene—^Δ^Ct of the calibrator for the target gene.

### 2.14. Statistical Analysis

The data are expressed as the mean ± standard deviation (SD). Statistical significance was determined while using an analysis of variance (ANOVA), followed by a multiple comparison test with Tukey’s multiple comparison test. *p* values ≤ 0.05 were considered to be significant. The statistical analyses were performed while using the GraphPad Prism 5 (GraphPad Software, CA, USA). 

## 3. Results

### 3.1. Tentative Characterization of SP Extract

The major peaks that were detected in *S. prunifolia* extract (Figure 1 and Table 1) were tentatively identified by comparison with previously literatures, as well as accurate mass and fragmentation pattern of mass spectra acquired in negative mode. The pseudeomolecular of caffeoyl glucoside was detected at *m/z* 341 and its fragment ion at *m/z* 179 ([caffeic acid (CA) − H]^−^) was produced by the loss of a glucosyl residue (162 Da) [25]. Thus, peak 1 was identified as isomer of caffeoyl glucoside. The fragmentation tendency of two peaks 8 and 13 was similar with that of dicaffeoyl glucoside isomers at *m/z* 503 based on these fragmentation pattern and previous literature [26], which showed a two predominant ions *m/z* 341 ([caffeoylglucoside − H]^−^) and 179 ([CA – H]^−^). Therefore, the peaks 8 and 13 were tentatively identified as dicaffeoyl glucoside isomers. *p*-Coumaroyl hexoside has a molecular weight of 326 and its predominant peak at *m/z* 325 [M − H]^−^ generated distinctive ion at *m/z* 163 due to the loss of a glucosyl residue [27]. Accordingly, *m/z* 325 of the peak 3 might be suggested as coumaroyl hexoside. Caffeoyl quinic acid (CQA) has a molecular weight of 354 and its various regional (3-, 4-, and 5-CQA) and geometrical (*cis* and *trans* configurations) isomers were reported in many natural products [25,27]. Peaks 2 and 4 displayed the characteristic mass fragment patterns of CQA isomer, including *m/z* 191 (quinic acid), 179, and 135 [25,27], which suggested that these peaks 2 and 4 may be CQA isomers. Quercetin and kaempferol flavonoids are representative derivatives that are found in many natural products [27,28,29]. Peak 6 was putatively identified as quercetin glucoside with *m/z* 463 and a characteristic fragment of quercetin at *m/z* 301 in negative mode, which was produced by eliminating the glucoside unit [28]. Peak 7 was showed a pseudomolecular ion of *m/z* 757, which experienced a hexoside loss (162 Da) to generate predominant ion at *m/z* 595 [29]. Furthermore, its mass fragment pattern provided evidence for the presence of quercetin aglycone at *m/z* 301 that was obtained from the loss of hexosyl-pentoside unit [29,30]. Thus, peak 7 was tentatively identified as quercetin hexosyl-pentoside hexoside. Peak 10, 15, and 17 were isomers showing a deprotonated ion [M − H]^−^ at *m/z* 799. Their main fragment ion pattern (*m/z* 637, 595, and 301) was similar to ion fragment pattern for peak 7, except for base peak at 637 *m/z*, to which the acetyl group (42 Da) was attached [31,32]. Accordingly, peak 10, 15, and 17 were tentatively identified as quercetin acetylhexoside-pentoside hexoside. In addition, the difference of 162 Da between peak 10 at *m/z* 799 and peak 11 at *m/z* 637 was attributed to loss of hexoside, and peak 11 was identified as quercetin acetylhexoside-pentoside [31,32]. Peak 20 was shown a pseudomolecular ion at *m/z* 813. Its major ion at *m/z* 637 [M – H − 176]^−^, 595, and 301, gave evidence of glucuronide (176 Da) residue loss and the mass fragment ion pattern of peak 20 was similar to that of peak 11 [33]. Therefore, peak 11 was tentatively identified as quercetin acetylhexoside-pentoside glucuronide. In addition, the kaempferol derivative was deduced by the main fragment ion at *m/z* 285. Peak 9 indicated a base peak at *m/z* 741, 579, 447, and 285 that was detected by the consecutive loss of hexoside (162 Da), pentoside (132 Da), and hexoside (162 Da), and it was tentatively identified as kaempferol hexosyl-pentoside hexoside [34]. Based on the mass ion fragment pattern of peak 9, peak 19 at *m/z* 783 showed a difference of 42 Da, corresponding to acetyl moiety (*m/z* 579 [M – H − hexoside-acetyl]^−^). Moreover, its mass fragment ions at *m/z* 621 and 285 indicated the loss of hexoside and presence of kaempferol aglycone. Thus, peak 19 was putatively identified as kaempferol acetylhexoside-pentoside hexoside [31,32].

### 3.2. SP Treatment Decreases Inflammatory Cells and Pro-Inflammatory Cytokine Contents in BALF

Mice that were treated with LPS exhibited a significant increase in the total cell count, as well as neutrophils, macrophages, and other inflammatory cells in the BALF compared with the normal control mice, as shown in Figure 2. However, SP-treated mice significantly reduced the number of inflammatory cells in the BALF when compared with the LPS-induced mice (Figure 2A). The protein content, TNF-α, IL-1β, and IL-6 levels in the BALF were significantly increased in the LPS-induced mice as compared with the normal controls. In contrast, the SP-treated mice significantly decreased protein content, TNF-α, IL-1β, and IL-6 levels in a dose-dependent manner as compared with the LPS-induced mice (Figure 2B–E).

### 3.3. SP Treatment Attenuates LPS-Induced Lung Histological Changes

We evaluated the effects of SP on LPS-induced lung histopathological changes (Figure 3). When compared with the normal control, the LPS-treated mice showed remarkable histopathological changes, as evidenced by the thickening of the alveolar wall, and the marked infiltration of inflammatory cells into the peribronchiolar and perivascular regions. The scores of the lesions also markedly increased as compared with the normal controls. However, pretreatment with SP resulted in less infiltration of inflammatory cells and reduced the thickening of alveolar wall in the lung tissues as compared with the LPS-treated mice. In the SP-treated mice, a meaningful reduction in the score of each lesion was also observed when compared with LPS-induced mice.

### 3.4. SP Treatment Suppresses LPS-Induced MAPKs, NF-κB Activation and Inflammatory Mediators in Lung Tissues

LPS-treated mice demonstrated a significant increase in the phosphorylation of JNK, Erk, and p38MAPK in the lung tissues when compared with normal control mice, as shown in Figure 4. The LPS-treated mice showed a significant increase in the phosphorylation of p65NF-κB as compared with the normal control. In addition, iNOS and TNF-α expression levels were markedly increased in the lung tissues of LPS-treated mice. However, pretreatment with SP significantly suppressed the phosphorylation of the MAPKs in the lung tissues in a dose-dependent manner when compared with the LPS-treated mice. The SP-treated mice showed a marked reduction of the phosphorylation of p65NF-κB and the expression levels of iNOS and TNF-α as compared with the LPS-treated mice.

### 3.5. SP Treatment Decreases Lipid Peroxidation and Induces Nrf2 Pathway and Antioxidant Enzymes in Lung Tissues

As presented in Figure 5, LPS-treated mice showed a significant decrease in the Nrf2 nuclear translocation in the lung tissues as compared with normal control mice. The HO-1 and NQO1 expression levels in the lung were slightly increased in the LPS-treated mice. Additionally, LPS-treated mice showed a significant elevation of TBARS levels and a significant decrease of GSH contents as compared with normal control. In contrast, SP treatment significantly increased the expression levels of Nrf2 in nuclear fraction when compared with the LPS-treated mice. The SP-treated mice also showed a significant increase of HO-1 and NQO1 levels in the lung tissues of LPS-treated mice. SP treatment also significantly decreased the TBARS levels, and increased GSH levels in lung tissues as compared with the LPS-treated mice.

### 3.6. SP Treatment Reduces Pro-Inflammatory Cytokine Production in TNF-α-stimulated NCI-H292 Cells

In this study, we used a nontoxic concentration of SP that was based on the results of the cytotoxicity analysis (Figure 6A). TNF-α treatment markedly increased the levels of TNF-α and IL-6 when compared with the non-stimulated cells, as shown in Figure 6B,C. However, SP-treated cells significantly decreased the TNF-α and IL-6 levels in a concentration-dependent manner when compared with the TNF-α-stimulated cells. TNF-α-stimulated cells exhibited a significant increase in the mRNA levels of TNF-α, IL-1β, and IL-6 in NCI-H292 cells. In contrast, SP treatment significantly reduced TNF-α, IL-1β, and IL-6 mRNA expression levels as compared with TNF-α-stimulated cells (Figure 6D–F).

### 3.7. SP Treatment Decreases Phosphorylation of MAPKs and NF-κB Activation in TNF-α-Stimulated NCI-H292 Cells

We investigated the effects of SP on MAPK signaling in TNF-α-stimulated cells (Figure 7). TNF-α treatment markedly increased the phosphorylation levels of Erk, JNK, and p38MAPK in NCI-H292 cells. TNF-α-stimulated cells exhibited a significant increase in the phosphorylation of p65NF-κB and TNF-α expression. However, treatment with SP significantly reduced the phosphorylation of Erk, JNK, and p38MAPK when compared with the TNF-α stimulated cells. Additionally, the phosphorylation of p65NF-κB and TNF-α expression levels were decreased in a concentration-dependent manner in the SP-treated cells.

### 3.8. SP Treatment Activates Nrf2 Pathways and Decreases Oxidative Stress, ROS Production and DPPH Radicals in TNF-α-stimulated NCI-H292 Cells

TNF-α treatment decreased the nuclear translocation of Nrf2, whereas HO-1 expression was increased in TNF-α-stimulated NCI-H292 cells, as shown in Figure 8. TNF-α-stimulated cells also exhibited a significant decrease in the GSH levels. However, treatment with SP significantly increased the translocation of Nrf2 into the nucleus with a concurrent increase of HO-1. The SP exhibited a significant decrease of ROS production and a significant increase of GSH and DPPH radical scavenging activity in a concentration-dependent manner. 

## 4. Discussion

In this study, we evaluated the anti-inflammatory effect of SP on TNF-α-stimulated NCI-H292 cells and an LPS-induced ALI mouse model. In TNF-α-stimulated NCI-H292 cells, SP pretreatment suppressed oxidative stress, as evidenced by scavenging DPPH radicals, suppressing ROS production, and activating Nrf2 nuclear translocation. SP pretreatment inhibited the production of pro-inflammatory cytokines/mediators and suppressed the phosphorylation of MAPKs and p65NF-κB. In the *in vivo* study, SP induced the activation of Nrf2 with an up-regulation of antioxidant enzymes, HO-1 and NQO1, and suppressed lipid peroxidation and restored the GSH contents in the lung tissues from LPS-treated mice. SP treatment effectively reduced the number of inflammatory cells, pro-inflammatory cytokines, including TNF-α, IL-1β, and IL-6, in BALF, and inflammatory cell infiltration into the lung tissues. Moreover, SP significantly suppressed LPS-induced MAPK and p65NF-κB activation the in the lung tissues.

LPS exposure not only promotes the release of inflammatory cytokines, but also induces the production of ROS, causing parallel tissue damage and aggravating inflammation [13,35]. Thus, LPS acts as an inducer of lung injury and it is used to establish a model of ALI [1,36]. In an experimental murine model of ALI, LPS caused damage to endothelial and epithelial barriers and activated neutrophils and macrophages, thereby releasing various pro-inflammatory cytokines, such as TNF-α, IL-1β, and IL-6 [37,38,39]. It has been reported that these pro-inflammatory cytokines activate and recruit neutrophils to cross the vascular endothelium into the lung and cause a cascading inflammatory response to amplify tissue injury [11,13,40]. In our study, SP treatment reduced the levels of TNF-α, IL-1β, and IL-6 in the BALF of the LPS-induced ALI model and down-regulated the TNF-α, IL-1β, and IL-6 mRNA expression levels in TNF-α-stimulated NCI-H292 cells. Consistent with these results, SP treatment significantly decreased the number of neutrophils and macrophages in the BALF and attenuated inflammatory infiltration into the lung tissues. Therefore, these findings indicate that SP treatment attenuates LPS-induced ALI via the downregulation of pro-inflammatory cytokines.

LPS can activate immune cells by binding to TLR4, thus triggering the TRIF- or MyD88-dependent pathways [41,42]. These pathways regulate the inflammatory response, thereby promoting the activation of the MAPKs and NF-κB. In the MAPK family, a series of protein kinases, such as JNK, Erk, and p38MAPK, are involved in diverse cellular responses, including inflammation-related pathways. These MAPKs are activated by either extracellular or intracellular signals [43,44], which in turn promotes early pro-inflammatory cytokine production during the inflammatory process [45,46,47]. The MAPK signaling pathway is known to be activated in response to LPS-induced ALI [11,48,49]. In our study, SP treatment effectively suppressed the phosphorylation of JNK, Erk, and p38MAPK in the LPS-induced ALI mouse model. Therefore, these results indicate that SP treatment effectively inhibited the inflammatory responses by blocking the activation of the MAPKs.

NF-κB is an upstream regulator of various inflammatory mediators/cytokines and it plays an important role in regulating the TLR4-mediated inflammatory response [50]. In the NF-κB signaling pathway, IκB kinase can phosphorylate IκB, and then translocate the p65-p50 dimer into the nucleus, where its binds to DNA to regulate the expression of pro-inflammatory genes, such as iNOS and TNF-α [43,47]. LPS can induce the release of pro-inflammatory cytokines through the activation of NF-κB signaling [51,52]. Accumulating evidence has demonstrated that the inhibition of NF-κB can attenuate LPS-induced ALI [11,43,53]. In this study, SP treatment markedly inhibited the phosphorylation of p65NF-κB and reduced pro-inflammatory mediators in both the TNF-α-stimulated NCI-H292 cells and LPS-induced ALI model. Thus, these findings suggest that SP might inhibit LPS-induced inflammatory responses in the lung via the suppression of NF-κB activation, resulting in the subsequent suppression of inflammatory mediators or cytokines.

Oxidative stress is characterized by the overproduction of ROS and it plays an important role in the pathogenesis of ALI [13]. ROS accumulation aggravates the inflammatory responses by promoting the expression of proinflammatory cytokines, and infiltration of inflammatory cells, leading to ALI [15,16]. In our study, SP treatment displayed a potent DPPH radical-scavenging activity and suppressed ROS generation in TNF-α-stimulated NCI-H292 cells. In addition, SP effectively attenuated lipid peroxidation and restored GSH concentration in the lung tissues of the LPS-induced ALI model and TNF-α-stimulated NCI-H292 cells. Thus, our findings indicate that SP has potent antioxidant properties that are likely to be responsible for its protective effect against acute lung damage that is caused by LPS treatment. Nrf2 plays an essential role in many inflammatory- and oxidative stress-related diseases, such as chronic obstructive pulmonary disease (COPD) and ALI [14,18]. Nrf2 protects a variety of tissue and cells against ROS through ARE-mediated induction of diverse antioxidant enzymes, including HO-1 and NQO1 [54]. Under the oxidative stress condition, Nrf2 is translocated from the cytosol to the nucleus, subsequently binding to ARE, which results in the expression of HO-1 and NQO1 [55]. It provides a host defense mechanism against oxidative stress and contributes to the anti-inflammatory activity [56,57]. Enhancing Nrf2 activation could attenuate the LPS-induced ALI [18]. In response to LPS-induced oxidative stress, the HO-1 and NQO1 levels were slightly increased in the LPS-treated THP-1 cells [58]. In *in vivo* studies, LPS treatment caused ROS-mediated oxidative stress that induced a compensatory increase of HO-1 and NQO1 in the LPS-induced ALI model [59,60]. Consistent with these results, LPS-treated mice showed a slight increase of HO-1 and NQO1 levels in lung tissues. In contrast, the pretreatment of SP induced the nuclear translocation of Nrf2. Moreover, we confirmed that SP pretreatment showed higher expression levels of HO-1 and NQO1 in lung tissue than those of LPS-treated mice. The phenomenon suggested that SP itself might promote the nuclear translocation of Nrf2 with an up-regulation of HO-1 and NQO1 (Figure 5 and Figure 8) and these results were consistent with results of previous studies [59,60]. Therefore, our results suggest that the protective effects of SP on the LPS-induced oxidative stress may be closely associated with the activation of Nrf2 and its potent antioxidant activities.

In the therapy of ALI, DEX is prescribed to prevent the progression of acute respiratory distress syndrome (ARDS, a severe form of ALI), according to the present patho-physiological concepts [61]. However, the prolonged use of DEX causes various adverse effects, including hyperglycemia, pancreatitis, infection, and gastrointestinal bleeding, which might be an important limiting factor. In addition, there is still no effective treatment for ALI [62]. The development of new agents is still urgently needed. Increasingly considerable attention has been given to the antioxidant and anti-inflammatory natural products, due to the significant therapeutic effect and relatively low toxicity of the herbal medicine [6,19]. *Spiraea* species have been reported to contain various diterpenes, diterpene alkaloids, terpenoid glycosides, and flavonoids [63]. The young leaves, fruits, and roots of SP have been used for the treatment of malaria, fever, and emesis in traditional medicine [21,22,64]. According to previous studies, the anti-oxidant properties of SP include scavenging DPPH radicals and superoxide anions and inhibiting NO production in LPS- or polymyristic acetate-stimulated RAW264.7 or murine microglia BV-2 cell lines [21,22]. In this study, caffeoyl quinic acid, quercetin, and kaempferol are the main identified active components of SP a. The caffeoyl quinic acid inhibited DPPH radical formation and reduced TBARS levels [65]. Caffeoyl quinic acid reduced TNF-α and IL-1β levels in carrageenan-induced rat paw edema model [66]. Quercetin decreased the levels of inflammatory cytokines and malondialdehyde contents, while increasing antioxidant enzyme activities (superoxide dismutase, glutathione peroxidase, and catalase) in LPS-induced airway inflammation and oxidative stress in the ALI rat model [67]. Kaempferol reduced myeloperoxidase activity and increased superoxide dismutase level. It also exhibited anti-inflammatory effects through inhibiting MAPKs and NF-κB pathways [45]. The pretreatment of methanol extract of SP leaves partially attenuated the inflammatory responses and oxidative stress by blocking the activation of NF-κB/MAPK pathways (Figure 4 and Figure 7) and by activating Nrf2 pathways (Figure 5 and Figure 8) in the LPS-induced ALI model and TNF-α-stimulated NCI-H292 cells, although SP could not completely restore the airway inflammation and oxidative stress that are induced by LPS or TNF-α. 

There are some limitations in this study. It is yet not known which components exerted anti-oxidants or anti-inflammatory activities. Thus, further study will be conducted to explore the differences in the activities or components of SP from other locational or seasonal conditions, and to investigate the major active components that exerted anti-oxidant activity in ALI model for therapeutic use of SP.

## 5. Conclusions

Overall, to our best knowledge, this is the first study investigating SP-mediated anti-inflammatory effects and anti-oxidant activities their underlying mechanisms in an *in vivo* model. We demonstrated that SP attenuated the inflammatory responses and ROS-mediated oxidative stress in an LPS-induced ALI mouse model and in TNF-α-stimulated NCI-H292 cells. Furthermore, these effects are considered to be associated with the inhibition of the MAPK/NF-κB pathways and the activation of Nrf2. Thus, our results suggest that SP has therapeutic potential for the treatment of inflammatory diseases.

## Figures and Tables

**Figure 1 antioxidants-09-00198-f001:**
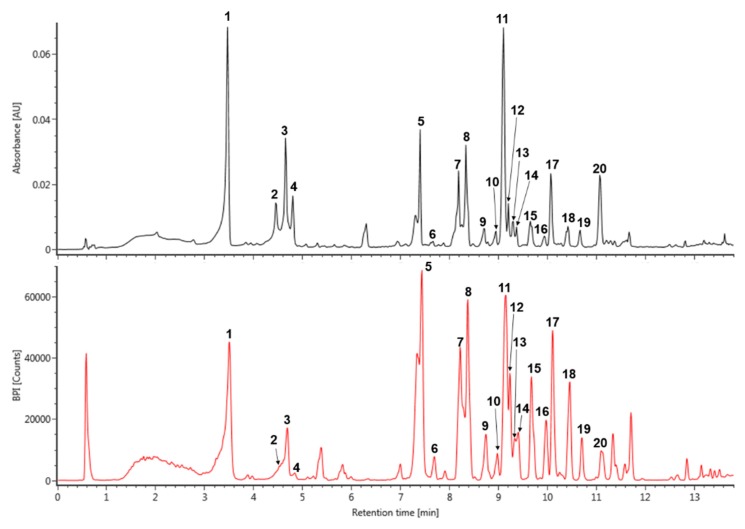
The UV (280 nm) and base peak intensity (BPI) chromatogram of *Spiraea prunifolia* var. *simpliciflora* in negative mode using ultra-performance liquid chromatography-quadrupole-time of flight mass spectrometry (UPLC-QToF-MS).

**Figure 2 antioxidants-09-00198-f002:**
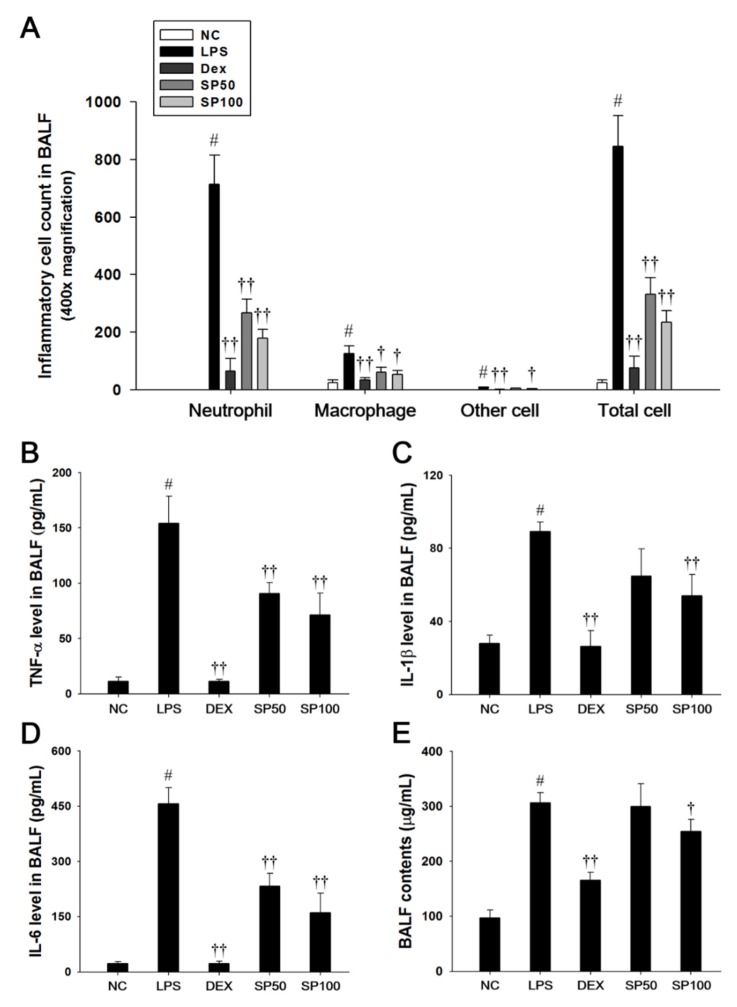
Effects of SP on inflammatory cell number, pro-inflammatory cytokines, and protein contents in the bronchoalveolar lavage fluid (BALF). (**A**) Inflammatory cells were counted by Diff-Quik stain reagent. (**B**) Tumor necrosis factor (TNF)-α, (**C**) interleukin (IL)-1β, and (**D**) IL-6 production in the BALF were determined by ELISA. (**E**) BALF protein contents were assessed by Bradford reagent. NC: normal control mice; LPS: LPS-induced mice; DEX: dexamethasone (3 mg/kg) + LPS-induced mice; SP: SP (50 or 100 mg/kg) + LPS-induced mice. The values are expressed as the means ± SD (*n* = 7/group). ^#^ Significantly different from NC group, *p* < 0.05; ^†, ††^ significantly different from LPS-treated group, *p* < 0.05 and *p* < 0.01.

**Figure 3 antioxidants-09-00198-f003:**
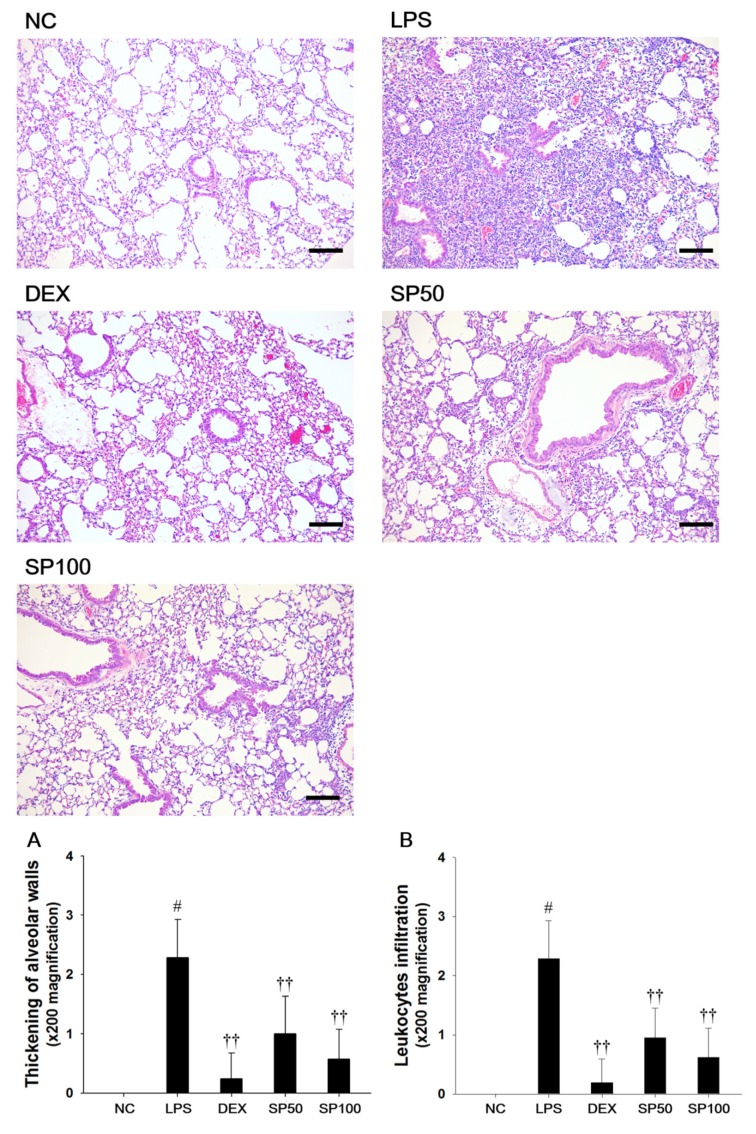
Effects of *Spiraea prunifolia* var. *simpliciflora* (SP) on airway inflammation in LPS-induced ALI mice. Histological examination of airway inflammation performed in the lung tissue by H&E staining. NC: normal control mice; LPS: LPS-induced mice; DEX: dexamethasone (3 mg/kg) + LPS-induced mice; SP 50: (50 mg/kg) + LPS-induced mice and SP 100: (100 mg/kg) + LPS-induced mice. Average pathologic scores of (**A**) thickening of alveolar walls and (**B**) leukocytes infiltration are expressed as the means ± SD (*n* = 7/group). ^#^ Significantly different from NC group, *p* < 0.05; ^†, ††^ significantly different from LPS-treated group, *p* < 0.05 and *p* < 0.01.

**Figure 4 antioxidants-09-00198-f004:**
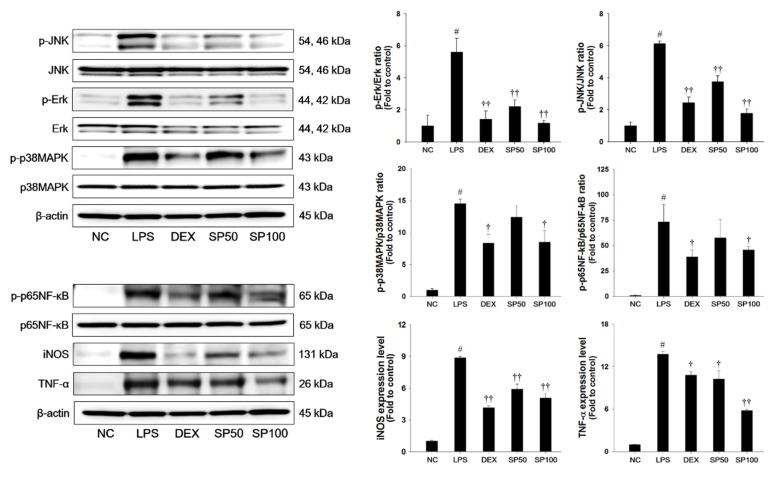
Effects of SP on activation of MAPKs, p65NF-κB, and inflammatory mediators in LPS-induced ALI mice. The protein levels of MAPKs (JNK, Erk, and p38MAPK), p65NF-κB, iNOS and TNF-α in the lung tissues were assessed by western blot analysis. β-actin was used to confirm equal protein loading. NC: normal control mice; LPS: LPS-induced mice; DEX: dexamethasone (3 mg/kg) + LPS-induced mice; SP 50: (50 mg/kg) + LPS-induced mice and SP 100: (100 mg/kg) + LPS-induced mice. The values are expressed as the means ± SD (*n* = 7/group). ^#^ Significantly different from NC group, *p* < 0.05; ^†, ††^ significantly different from LPS-treated group, *p* < 0.05 and *p* < 0.01.

**Figure 5 antioxidants-09-00198-f005:**
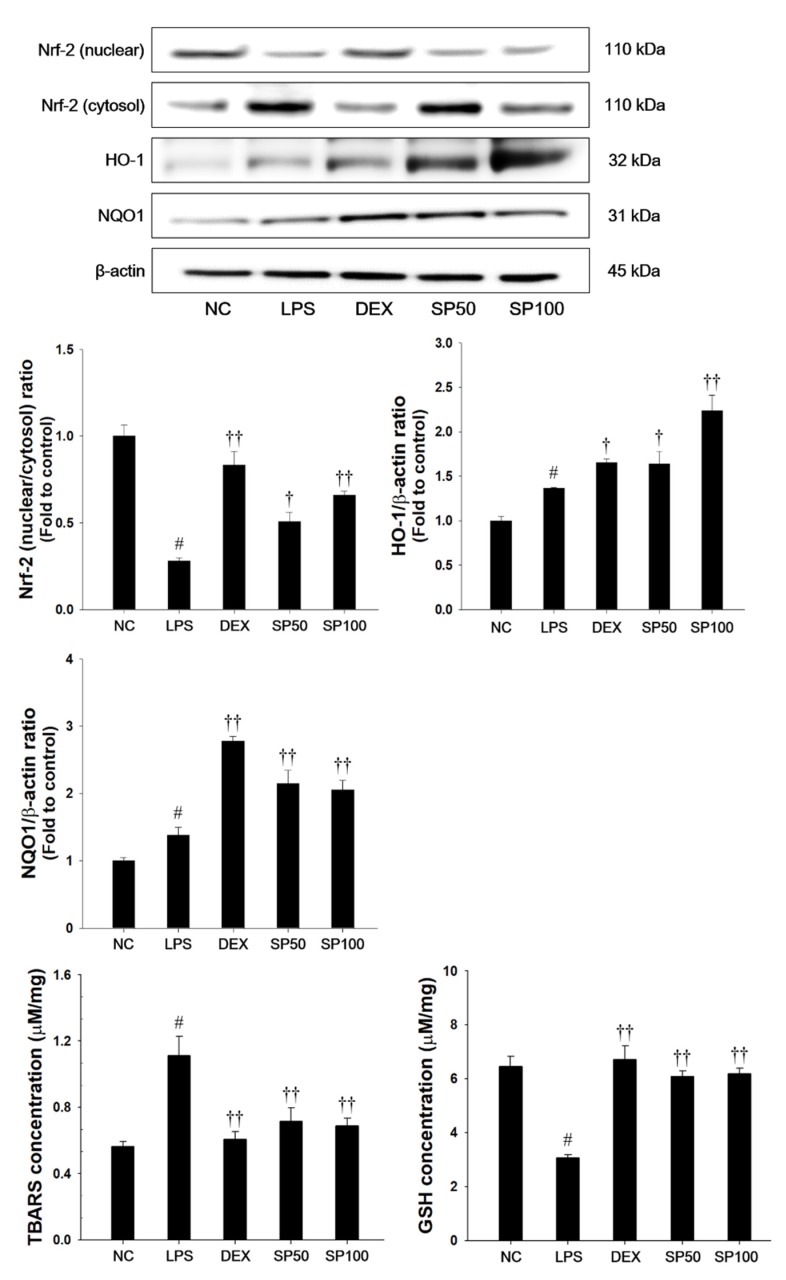
Effects of SP on Nrf2 pathways, lipid peroxidation and glutathione (GSH) in LPS-induced ALI mice. The protein levels of Nrf-2, HO-1, and NQO1 in the lung tissues were assessed by western blot analysis. β-actin was used to confirm equal protein loading. The levels of TBARS and GSH were measured in the lung tissues. NC: normal control mice; LPS: LPS-induced mice; DEX: dexamethasone (3 mg/kg) + LPS-induced mice; SP 50: (50 mg/kg) + LPS-induced mice and SP 100: (100 mg/kg) + LPS-induced mice. The values are expressed as the means ± SD (*n* = 7/group). ^#^ Significantly different from NC group, *p* < 0.05; ^†, ††^ significantly different from LPS-treated group, *p* < 0.05 and *p* < 0.01.

**Figure 6 antioxidants-09-00198-f006:**
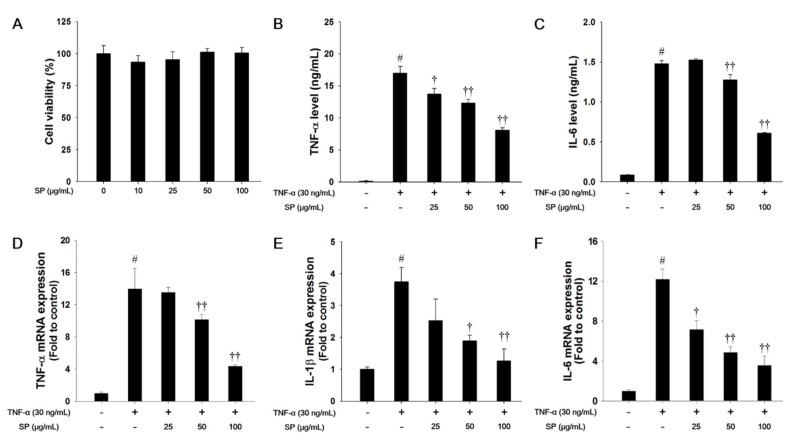
Effects of SP on cell viability and pro-inflammatory cytokines in TNF-α-stimulated NCI-H292 cells. (**A**) Cell viability was measured using a WST-1 and SP were treated with 10, 25, 50, and 100 μg/mL for 24 h. The (**B**) TNF-α and (**C**) IL-6 production in TNF-α-stimulated NCI-H292 cells were determined by ELISA. The culture medium were changed 0.1% FBS media and treated with SP (25, 50, 100 μg/mL) for 1 h and incubated with TNF-α (30 ng/mL) for 30 min. (TNF-α) or 24 h (IL-6), respectively. The mRNA levels of (**D**) TNF-α, (**E**) IL-1β, and (**F**) IL-6 mRNA levels in TNF-α-stimulated NCI-H292 cells. The cells were treated with SP (25, 50, 100 μg/mL) for 2 h and incubated with TNF-α (30 ng/mL) for 18 h. The values are expressed as the means ± SD (*n* = 3). ^#^ Significantly different from control, *p* < 0.05; ^†, ††^ significantly different from TNF-α-treated group, *p* < 0.05 and *p* < 0.01.

**Figure 7 antioxidants-09-00198-f007:**
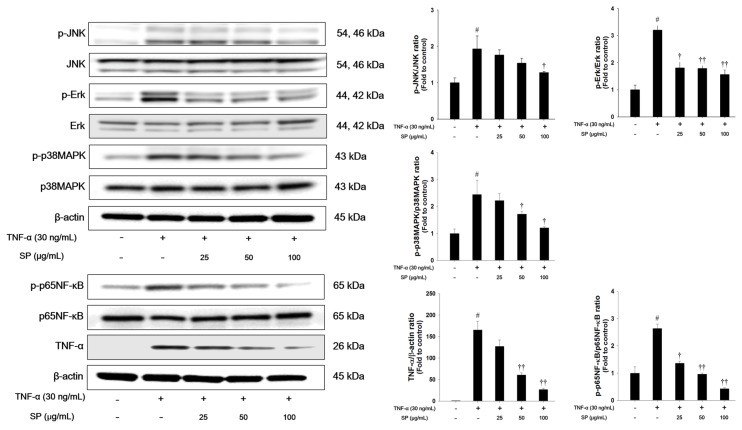
Effects of SP on activation of MAPKs, p65NF-κB, and TNF-α in TNF-α-stimulated NCI-H292 cells. The protein levels of MAPKs, p65NF-κB, and TNF-α expression were assessed by western blot analysis. Cells were changed 0.1% FBS media for 2 h and treated with SP (25, 50, 100 μg/mL) for 1 h and incubated with TNF-α (30 ng/mL) for 30 min. (MAPKs), 2 h (p65NF-κB), and 24 h (TNF-α), respectively. β-actin was used to confirm equal protein loading. The values are expressed as the means ± SD (*n* = 3). ^#^ Significantly different from control, *p* < 0.05; ^†, ††^ significantly different from TNF-α-treated group, *p* < 0.05 and *p* < 0.01.

**Figure 8 antioxidants-09-00198-f008:**
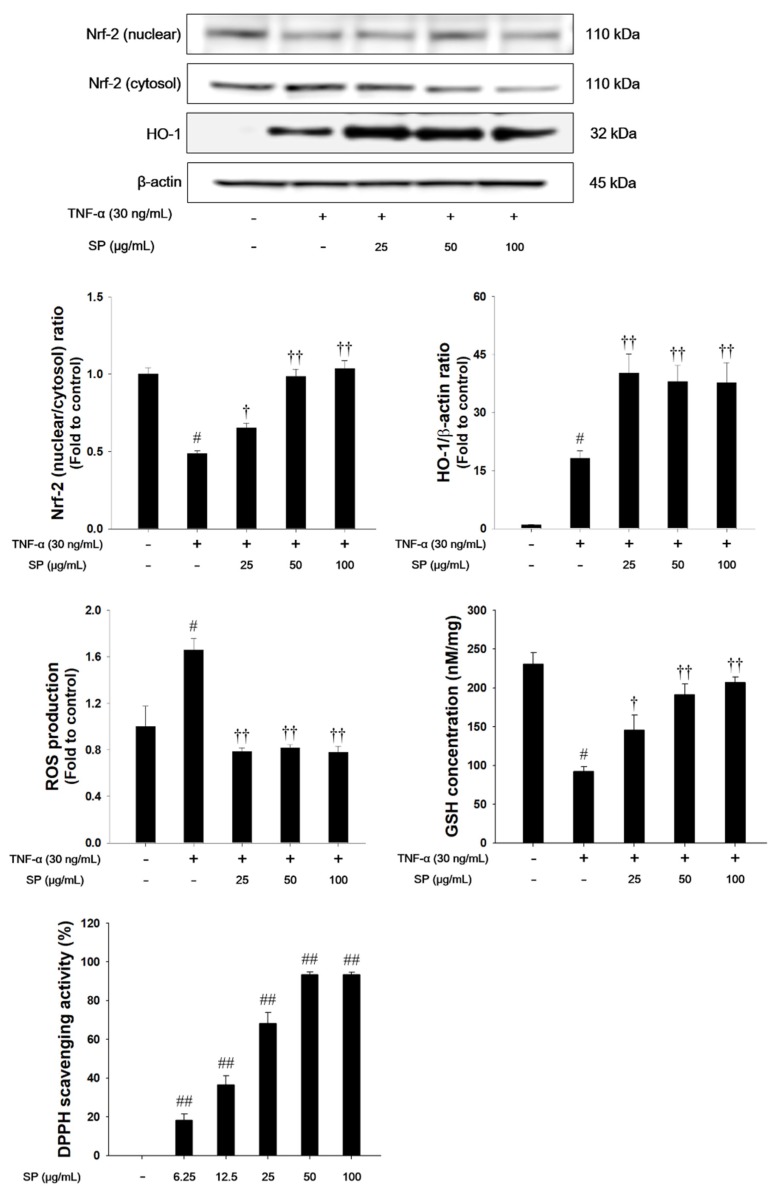
The effects of SP on Nrf-2 pathways, ROS production, GSH and 2,2-Diphenyl-1-Picryl Hydrazyl (DPPH) radicals in TNF-α-stimulated NCI-H292 cells. The protein levels of Nrf-2 and HO-1 expression were assessed by western blot analysis. β-actin was used to confirm equal protein loading. The ROS production and GSH were measured in TNF-α-stimulated NCI-H292 cells. Cells were changed 0.1% FBS media for 2 h and treated with SP (25, 50, 100 μg/mL) for 1 h and incubated with TNF-α (30 ng/mL) for 1 h (Nrf-2 and HO-1), and 24 h (ROS production and GSH). The DPPH radical scavenging activity was evaluated by DPPH assay. The values are expressed as the means ± SD (n = 3). ^#,##^ Significantly different from control, *p* < 0.05 and *p* < 0.01; ^†, ††^ significantly different from the TNF-α-treated group, *p* < 0.05 and *p* < 0.01.

**Table 1 antioxidants-09-00198-t001:** Tentatively identification of major peaks detected in *Spiraea prunifolia* var. simpliciflora.

No.	t*_R_* (min)	Detected *m/z* [M − H]^−^	Exacted *m/z* [M − H]^−^	Error (ppm)	Formula	Fragments	Identification
1	3.51	341.0879	341.0878	0.19	C_15_H_18_O_9_	179	caffeoyl glucoside
2	4.49	353.0866	353.0878	−0.81	C_16_H_18_O_9_	191, 179, 135	caffeoyl quinic acid
3	4.69	325.0921	325.0929	−0.65	C_15_H_18_O_8_	163, 145, 135	coumaroyl hexoside
4	4.80	353.0866	353.0878	−0.61	C_16_H_18_O_9_	191, 179, 135	peak 2 isomer
5	7.44	439.1247	439.1246	−0.76	C_20_H_24_O_11_	341, 179, 161	unknown
6	7.69	463.0893	463.0882	0.12	C_21_H_20_O_12_	301, 271, 255	quercetin glucoside
7	8.21	757.1626	757.1622	0.66	C_35_H_34_O_19_	595, 301, 271	quercetin hexosylpentoside-hexoside
8	8.37	503.1180	503.1195	−0.26	C_24_H_24_O_12_	341, 179, 161	dicaffeoyl glucoside
9	8.74	741.1682	741.1672	−0.17	C_35_H_34_O_18_	579, 455, 285	kaempferol hexosylpentoside-hexoside
10	8.97	799.1735	799.1727	−0.10	C_37_H_36_O_20_	637, 595, 301	quercetin acetylhexoside-pentoside hexoside
11	9.14	637.1417	637.1410	−0.06	C_28_H_30_O_17_	595, 335, 301,	quercetin acetylhexoside-pentoside
12	9.23	617.1504	617.1512	0.43	C_29_H_30_O_15_	455, 395, 179,	unknown
13	9.32	503.1208	503.1195	−0.44	C_24_H_24_O_12_	341, 179, 161	peak 8 isomer
14	9.41	487.1235	487.1246	−0.36	C_24_H_24_O_11_	323, 161	unknown
15	9.71	799.1735	799.1727	−1.17	C_37_H_36_O_20_	637, 301	peak 10 isomer
16	9.97	307.0825	307.0823	0.60	C_15_H_16_O_7_	285, 161	unknown
17	10.11	799.1735	799.1727	1.18	C_37_H_36_O_20_	637, 301	peak 10 isomer
18	10.46	601.1573	601.1563	0.23	C_29_H_30_O_14_	439, 179, 135	unknown
19	10.71	783.1786	783.1778	0.01	C_37_H_36_O_19_	621, 285	kaempferol acetylhexoside-pentoside hexoside
20	11.09	813.1893	813.1884	0.43	C_38_H_38_O_20_	637, 595, 301	quercetin acetylhexoside-pentoside glucuronide

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
