# Peer review of "Spiraea prunifolia var. simpliciflora Attenuates Oxidative Stress and Inflammatory Responses in a Murine Model of Lipopolysaccharide-Induced Acute Lung Injury and TNF-α-Stimulated NCI-H292 Cells"

_antioxidants, 2020, doi:10.3390/antiox9030198_

Round 1

Reviewer 1 Report

The authors evaluated the effects of a methanol extract of SP leaves on inflammation in TNF-α-stimulated human airway epithelial (NCI-H292) cells and in an LPS-induced ALI mouse model. They have shown many results, clearly articulated the mechanism of action of SP, and have well written discussions. It was a very interesting research.

One major concern is that SP is a methanol extract. As a matter of course, the extract contains many components as shown in Fig. 1. Therefore, it is completely unknown which component exhibited the antioxidant effect shown in this study. However, this manuscript does not consider any of these important points, the results of Fig. 1 (Of course, there are related descriptions in L474 to L483).

This is important for utilizing this methanol extract as a therapeutic drug for ALI. In particular, these components are likely to be significantly affected by the producing location of the SP and the climate, and greatly affect its biological activity. The mechanism of action of the biological activity of SP is also very important, but these points must also be fully discussed in the section of Discussion.

Author Response

We really agreed with your comments. The components of natural products showed the variation by its production area’s location and climate (season). It has been described in the Discussion section as “limitation of this study”. As you pointed, further study will be conducted to explore the activities and components of SP from other locations or seasons, and to investigate the main active components which exhibit the antioxidant effect in ALI model and NCI-H292 cell to therapeutic utilization of SP.

The response to the reviewer's comments as follow:

In this study, the identified active components of SP are mainly caffeoyl quinic acid, quercetin and kaempferol. The caffeoyl quinic acid inhibited DPPH radical formation and reduced TBARS levels [67]. Caffeoyl quinic acid reduced TNF-α and IL-1β levels in carrageenan-induced rat paw oedema model [68]. Quercetin decreased levels of inflammatory cytokines and malondialdehyde contents, whereas increased antioxidant enzyme activities (superoxide dismutase, glutathione peroxidase, and catalase) in LPS-induced airway inflammation and oxidative stress in ALI rat model [69]. Kaempferol reduced myeloperoxidase activity and increased superoxide dismutase level. It also exhibited anti-inflammatory effects through inhibiting of MAPKs and NF-κB pathways [45].

Additionally, there are some limitations in this study. It is yet not known which components exerted anti-oxidants or anti-inflammatory activities. Thus, further study will be conducted to explore the differences in activities or components of SP from other locational or seasonal conditions, and to investigate the major active components which exerted anti-oxidant activity in ALI model for therapeutic use of SP.

Reviewer 2 Report

The authors conducted a thorough study of the characterization and anti-inflammatory effects of Spiraea prunifolia var. simpliciflora (SP) in both an in vivo mouse ALI model and TNFa-stimulated lung cell culture (NCI-H292) model. The methods are thoroughly described. The results are mostly consistent and strongly indicate the anti-inflammatory effect of SP in appropriate models, as hypothesized. However, the authors need to clarify sample size / number of replicates and some of the results and controls further.

Major comments:

Methods: for each method, please include number of mice / treatment group or number of replicates (cell culture). Need to discuss use of dexamethasone (DEX) as a positive control – it works better than SP in almost all cases but is not discussed in the text. Please discuss why SP would be an important treatment option if DEX works better. Methods, page 3, lines 91-93 are unclear. What were the treatment groups? How many mice in each group? How were mice divided by cage into each group (were they all housed in the same cage by group or randomized)? Importantly: was the DEX administered in addition to the SP (in which case the results show SP attenuating the affect) or as a separate group (as a positive control group)? Figure 5 / text section 3.5 do NOT agree. The text (page 10, lines 343-444) states that HO-1 and NQO1 expression was decreased in LPS-treatment; however, Figure 5 shows that both HO-1 and NQO1 increased both following LPS and in all treatment groups. Making DEX/SP not a rescue, but an exacerbation compared to the negative control and LPS treatment groups. Please correct in text and add to discussion. Figure 8 / text section 3.8 do NOT agree. The text (page 13, lines 395-396) states that TNFa treatment decreased HO-1. However, figure 8 shows that TNFa increased HO-1, which was only further increased by DEX/SP treatment compared to TNFa-stimulated and negative control groups. Please correct in text and add to discussion.

Minor comments:

Figure 5 lists MDA, which is not shown (page 11, line 354). Please clarify why only male mice were used. Please clarify case grouping relative to treatment grouping. Please add to the discussion the fact that SP is only a partial rescue under almost all conditions (does not fully restore to baseline).

Author Response

Major comments:

1. Methods: for each method, please include number of mice / treatment group or number of replicates (cell culture).

It has been revised in Materials and Methods section. Number of mice/treatment group and number of replicates were represented in the text for each experiment condition.

2. Need to discuss use of dexamethasone (DEX) as a positive control – it works better than SP in almost all cases but is not discussed in the text. Please discuss why SP would be an important treatment option if DEX works better.

It has been described in the Discussion section as follow: In the therapy of ALI/ARDS, DEX is prescribed to prevent progression of ARDS according to present patho-physiological concepts [61]. However, prolonged use of DEX caused various adverse effects, including hyperglycemia, pancreatitis, infection, and gastrointestinal bleeding, which may be important limiting factor. In addition, there is still no effective treatment for ALI [62]. The development of new agents is still urgently needed. Due to significant therapeutic effect and relatively low toxicity of the herbal medicine, increasingly considerable attention has been given to the antioxidant and anti-inflammatory natural products [63,64].

3. Methods, page 3, lines 91-93 are unclear. What were the treatment groups? How many mice in each group? How were mice divided by cage into each group (were they all housed in the same cage by group or randomized)? Importantly: was the DEX administered in addition to the SP (in which case the results show SP attenuating the affect) or as a separate group (as a positive control group)?

It has been described a detailed grouping information in the Materials and Methods section as as you pointed out. In addition, DEX was treated to mice of DEX group only (a separate and positive group). The SP (50 or 100 mg/kg) was dissolved in distilled water with 2% dimethyl sulfoxide (DMSO; Sigma-Aldrich, St. Louis, MO, USA) and dexamethasone (DEX 3 mg/kg) were dissolved in distilled water before treatment daily. SP and DEX was administered orally from day 0 to day 5. DEX was used as a positive control [7]. Mice were treated with LPS (form Escherichia coli O111:B4; Sigma-Aldrich) 20 μg in 50 μl phosphate-buffered saline (PBS; Gibco, San Diego, CA, USA) by intranasal (i.n.) instillation 1 h after SP and DEX treatment on day 3. The NC group were treated with vehicle (2% DMSO) and given 50 μl PBS only by i.n. instillation on day 3. Total of 35 male mice were randomly divided into the control and 4 treatment groups. The animals were housed three or four per cage and each group consisted of 7 mice.  Normal control (NC) group: treated with vehicle (2% DMSO) from day 0 to day 5 and given 50 μl PBS without LPS on day 3 LPS group: treated with vehicle (2% DMSO) from day 0 to day 5 and given LPS 20 μg in 50 μl PBS on day 3 DEX group: treated with DEX 3 mg/kg only from day 0 to day 5 and given LPS 20 μg in 50 μl PBS on day 3 SP50 group: treated with SP 50 mg/kg only and given LPS 20 μg in 50 μl PBS on day 3 SP100 group: treated with SP 100 mg/kg only and given LPS 20 μg in 50 μl PBS on day 3.

4. Figure 5 / text section 3.5 do NOT agree. The text (page 10, lines 343-444) states that HO-1 and NQO1 expression was decreased in LPS-treatment; however, Figure 5 shows that both HO-1 and NQO1 increased both following LPS and in all treatment groups. Making DEX/SP not a rescue, but an exacerbation compared to the negative control and LPS treatment groups. Please correct in text and add to discussion.

As you pointed out, it has been revised in Results section and described in the Discussion section as follow: Results section: HO-1 and NQO1 expression levels in lung were slightly increased decreased in the LPS-treated mice. Discussion section: Nrf2 protects a variety of tissue and cells against ROS through ARE-mediated induction of diverse antioxidant enzymes, including HO-1 and NQO1 [54]. Under oxidative stress condition, Nrf2 is translocated from the cytosol to the nucleus, subsequently binds to ARE, resulting in expression of HO-1 and NQO1 [55]. It provides a host defense mechanism that oxidative stress and contributes to the anti-inflammatory activity [56,57]. Enhancing Nrf2 activation could attenuate the LPS-induced ALI [18]. In response to LPS-induced oxidative stress, HO-1 and NQO1 levels were slightly increased in the LPS-treated THP-1 cells [58]. In in vivo studies, LPS treatment caused ROS-mediated oxidative stress that induced a compensatory increase of HO-1 and NQO1 in LPS-induced ALI model [59,60]. Consistent with these results, LPS-treated mice showed a slight increase of HO-1 and NQO1 levels in lung tissues. In contrast, pretreatment of SP induced the nucleus translocation of Nrf2. Moreover, we confirmed that SP pretreatment showed a higher expression levels of HO-1 and NQO1 in lung tissue than those of LPS-treated mice. The phenomenon suggested that SP itself may promote the nuclear translocation of Nrf2 with up-regulation of HO-1 and NQO1 and these results were consistent with results of previous studies [59,60].

5. Figure 8 / text section 3.8 do NOT agree. The text (page 13, lines 395-396) states that TNFa treatment decreased HO-1. However, figure 8 shows that TNFa increased HO-1, which was only further increased by DEX/SP treatment compared to TNFa-stimulated and negative control groups. Please correct in text and add to discussion.

It has been corrected in Results section and discussed in the Discussion section as above 4. Results section: TNF-α treatment decreased the nuclear translocation of Nrf2, whereas HO-1 expression was increased in TNF-α-stimulated NCI-H292 cells.  Discussion section: described in 4.

Minor comments:

1. Figure 5 lists MDA, which is not shown (page 11, line 354).

It has been revised “MDA” to “TBARS” in legend of Figure 5. The levels of TBARS and GSH were measured in the lung tissues.

2. Please clarify why only male mice were used.

It has been clarified in the Materials and Methods section as follow: Materials and Methods section: It has been reported that male mice were more susceptible to LPS-induced airway inflammation compared to female mice [24]. Thus, we used only male mice in LPS-induced ALI model. 

3. Please clarify case grouping relative to treatment grouping.

The grouping information has been clarified in the Materials and Methods section as described above.

4. Please add to the discussion the fact that SP is only a partial rescue under almost all conditions (does not fully restore to baseline).

It has been add and discuss the fact that the partial ameliorative effects of SP on LPS-induced ALI model as you pointed out. Although SP could not completely restore the airway inflammation and oxidative stress induced by LPS or TNF-α, the pretreatment of methanol extract of SP leaves partially attenuated the inflammatory responses and oxidative stress by blocking the activation of NF-κB/MAPK pathways and by activating Nrf2 pathways in the LPS-induced ALI model and TNF-α-stimulated NCI-H292 cells.

Round 2

Reviewer 1 Report

The figure number corresponding to the discussion description should be inserted; L500~L503, L527~L529.

Author Response

Reviewer's comments

1. The figure number corresponding to the discussion description should be inserted; L500~L503, L527~L529.

As you pointed out, it has been inserted figure numbers in L500~L503 and L527~L529 as follow:

  • The phenomenon suggested that SP itself may promote the nuclear translocation of Nrf2 with up-regulation of HO-1 and NQO1 (Figure 5 and 8)

  • The pretreatment of methanol extract of SP leaves partially attenuated the inflammatory responses and oxidative stress by blocking the activation of NF-κB/MAPK pathways (Figure 4 and 7) and by activating Nrf2 pathways (Figure 5 and 8) in the LPS-induced ALI model and TNF-α-stimulated NCI-H292 cells.